# Effect of Heat Treatment on Microstructure and Properties of 24CrNiMo Alloy Steel Formed by Selective Laser Melting (SLM)

**DOI:** 10.3390/ma14030631

**Published:** 2021-01-29

**Authors:** Yongsheng Zhao, Chenggang Ding

**Affiliations:** School of Materials Science and Engineering, Dalian Jiaotong University, Dalian 116028, China

**Keywords:** 24CrNiMo alloy steel powder, SLM, heat treatment, thermal fatigue, grain orientation

## Abstract

The 24CrNiMo alloy steel powder was used as experimental material. The microstructure and mechanical properties of as-deposited, quenched and tempered (QT) and stress-relief annealed (SR) specimens were analyzed. The X-ray diffraction (XRD) analysis showed that the specimens in the three states were mainly ferrite (α-Fe), in which the as-deposited samples had puny face-centered cubic (FCC) structure Fe (γ-Fe). The microstructure observation showed that the as-deposited specimens were made up of ferrite, granular bainite, a small amount of cementite and residual austenite. The tensile test results indicated that the tensile strength and yield strength of the as-deposited specimens were 1199 MPa and 1053 MPa respectively, and the elongation at break was 10.7%. The elongation of QT and SR specimens increased to 11.6% and 12.8%, respectively. The electron backscattered scattering detection (EBSD) analysis results showed that the small-angle grain boundary content of the as-deposited samples was 58%, and large-angle grain boundary content was 15%. After QT and SR, small-angle and large-angle grain boundaries were obtained than those in the as-deposited specimens. The high-temperature friction and wear properties and thermal fatigue performances of the QT and SR specimens were improved significantly. The QT specimens had the smallest wear and thermal fatigue crack lengths, excellent resistance to friction and wear performance and prevention of crack growth, with an ideal comprehensive properties.

## 1. Introduction

Cr-Ni low-alloy structural steels are widely used in the manufacture of critical internal parts of industrial equipment due to their dramatic advantages of remarkable machinability and mechanical properties, such as 12CrNi2 for camshafts of emergency diesel engines [1,2,3,4] and 34CrNiMo6 for ultra-high-strength fasteners [5]. At present, 24CrNiMo alloy steel is used as the key material of high-speed rail brake discs [6]. Because of its harsh service conditions, the comprehensive performance and service life of the equipment are gravely restricted. For example, the high-temperature friction and wear and thermal fatigue performance requirements are extremely high [7], and the processing difficulty is very complicated. Selective laser melting (SLM) is a new type of rapid laser manufacturing technology which is based on the principle of layered stacking to achieve the rapid manufacturing of metal parts. The density of the specimen can approach 100%, and the mechanical properties are equivalent to those of the forging process [8,9]. Compared with traditional manufacturing methods, SLM technology has obvious advantages [10,11,12] and can be utilized to manufacture metal parts directly from raw materials [13]. Chen et al. [14,15] of Northeastern University have carried out systematic research on the additive manufacturing of 24CrNiMo alloy steel. The results show that 24CrNiMo alloy steel brake discs with superior performance prepared by SLM have great potential application value. The processing problem can be well solved by using the laser material additive manufacturing method. After heat treatment [16,17], mechanical properties such as high-temperature friction and wear properties and thermal fatigue properties of the parts are improved. The SLM technology is of immense significance in the manufacture of brake discs for high-speed trains. Some scholars [18] fabricated Ti6Al4V samples using SLM technology and heat-treatment under vacuum, which changed the grain size, improved the tensile properties and enhanced the fatigue resistance. The fatigue behavior of SLM-formed Co–Cr–Mo–W alloy has been studied by Xin Dong et al. [19]. The results showed that the fatigue life of SLM-formed Co–Cr–Mo–W alloy is longer than that of cast Co–Cr–Mo–W alloy. In previous research [20], the effects of SLM process parameters on the microstructure and properties of 24CrNiMo low alloy steel were studied. However, there are few reports on the relationship between microstructure characteristics such as grain size, grain orientation, grain boundary quantity and high-temperature friction and wear properties and thermal fatigue properties. The research work focused on the influences of laser energy density and different scanning strategies on the formability of 24CrNiMo specimens prepared by SLM [21]. Therefore, the research on heat treatment processing of QT and SR are insufficient at present. Against this background, 24CrNiMo alloy steel specimens were prepared by SLM technology under the optimized process parameters in this paper. The relationship between microstructure characteristics such as grain size, grain orientation, grain boundary quantity and high-temperature friction and wear properties and thermal fatigue properties of as-deposited QT and SR states were studied. This provides the test basis for the preparation of the metal parts of the brake discs of high-speed trains and the safe operation of high-speed rail vehicles.

## 2. Materials and Methods

### 2.1. Test Material

In this paper, the experimental material was 24CrNiMo low-alloy steel powder with a particle size of 15 to 53 μm. The particle size distribution was as follows: D10 = 20.8 μm, D50 = 35.0 μm, D90 = 57.2 μm, Dav = 37.67 μm. The substrate material was a 30CrNiMo steel with a size of 250 × 250 × 30 mm^3^. The chemical composition of 30CrNiMo and 24CrNiMo is indicated in Table 1.

### 2.2. Test Method

#### 2.2.1. Preparation of Deposited Sample and Heat Treatment Method

In the experiment, an EP-M250 metal 3D printer (Beijing Yijia 3D Technology Co., Ltd, Beijing, China) was used to control the additive manufacturing process. The optimized process parameters were: laser power, P = 300 W; laser scanning speed, V = 550 mm/s; scanning distance, d = 0.11 mm, fixed layer thickness, h = 50 μm; substrate preheating temperature, 50 °C; and the shielding gas was argon (gas flow rate of 45 m^3^/h). The heat treatment process is shown in Table 2.

#### 2.2.2. Test Method for Microstructure

After pre-grinding and polishing, the metallographic specimens wasere corroded by 2% HNO3 alcohol solution for 15 s. Then, the microstructure of the deposited layers was observed using a LeicaDMi8A (Leica microsystems Ltd., Wetzlar, Germany) metallographic microscope. The phase of the additive layers was analyzed by X-ray diffraction (XRD, PANalytical, Almelo, The Netherlands), and then the micro-morphology of the specimens was photographed by Zeiss SUPRA-55 (Carl Zeiss GmbH, Jena, Germany) field emission scanning electron microscope (SEM). After shooting, an EBSD probe was used for EBSD scanning of the specimen, and HKL-Channel 5 ( Oxford Instruments, Oxon, UK, Version 3.0) software was utilized to analyze the grain size, grain orientation and misorientation of the specimens.

#### 2.2.3. Test Method for Mechanical Properties

The density of the specimens was observed using a DX-100E (Qunlong Instrument Co., Ltd., Xiamen, China) automatic electronic densitometer, and the microhardness of the additive specimens from the near-base layer to the surface layer was measured using an FM-700 (Future, Tokyo, Japan) microhardness tester with a load of 200 gf. The test location was from the surface layer to the inner layer, and the test interval was 0.25 mm. Then, at the same layer of each test point, the hardness value was tested twice at the interval of 2 mm. Finally, average value of the three points was taken as the hardness value of the position. A DNS300 (China Machinery Test equipment Co., Ltd., Jilin, China) electronic universal testing machine was used for a tensile test on the specimens. The high-temperature friction and wear properties of each test specimen were tested using an MM-5G (Jide Machinery and equipment Co., Ltd., Jinan, China) screen explicit high-temperature wear tester. Thermal fatigue properties of the specimens were tested by a self-made metal thermal fatigue testing machine made in the laboratory. The specimen size is shown in Figure 1 and the test parameters are given in Table 3.

## 3. Test Results and Analysis

### 3.1. Results and Analysis of Microstructure

#### 3.1.1. Phase Analysis

Figure 2 shows the XRD diffraction peaks of the three states of test specimens. It can be seen that there was no significant difference in the phase of each test specimen, which was composed of body-centered cubic (BCC) structure (α-Fe). There were also some faint diffraction peaks of face-centered cubic (FCC) structure Fe (γ-Fe) in the as-deposited state and SR state. However, the peak value became higher after heat treatment, indicating that the crystallinity was improved. The strength of the as-deposited specimens were 1136 MPa, and the strength of the QT specimens and the SR specimens were 1013 MPa and 1047 MPa, respectively.

#### 3.1.2. Microstructure Analysis

The as-deposited specimens of 24CrNiMo alloy steel were prepared by selective laser melting (SLM) with optimized process parameters. The microstructure of as-deposited specimens is indicated in Figure 3a, and the density of the specimens approached 99.78%. Some researchers have obtained granular bainite, lath martensite and carbon-rich retained austenite [20] in the specimens prepared by SLM. Due to the forming characteristics of SLM, the martensitic structure is easily formed in the alloy steel in the molten pool. Under the action of reheating of the subsequent additive layer, the martensite will be tempered to a certain extent. The area near the bottom of the new additive layer is most obviously affected by heat and is heated to a higher temperature. After continuous rapid cooling, granular bainite appeared in the interior. When the laser beam was scanned in the powder bed, a sunken molten pool was formed. The area with the highest temperature was the bottom of the sunken molten pool [22]. Martensite in the nearest cladding layer below was heated above the austenitizing temperature and started to transform into austenite. The austenite nucleated and grew preferentially in the carbon-rich regions such as grain boundaries and lath martensite boundaries. At the same time, the carbon element in the surrounding martensite diffused into the austenite, which improved the stability of the austenite and reduced the Ms point [23]. The phase transformation was difficult to achieve in the cooling process, and the retained austenite was formed. Because the time for the HAZ to be heated above the austenitizing temperature was quite short, there was just a small amount of austenite. The carbon content in martensite decreased obviously owing to the diffusion of carbon element in martensite into austenite during heating. As a result, the precipitation of carbides in the heat-affected zone decreased obviously. The microstructure of the QT samples is given in Figure 3b. After QT treatment, recovery recrystallization formed tempered sorbite structure, and retained austenite decomposed obviously. As shown in Figure 3c, the microstructure of the specimens had no obvious change compared with the as-deposited state after SR treatment.

### 3.2. EBSD Results and Analysis

The grain orientation was determined according to the BC diagram (mass imaging map). Grain orientation is the relationship between a grain coordinate system and specimen coordinate system, which can be described by the Euler angle. The Euler angular distribution of the three states of grains is given in Figure 4. It was found that the colors of the adjacent ferrites were similar in the Euler angular orientation reconstruction. However, there was a great difference in color between the two types of ferrite. This shows that the crystal phase angle difference between ferrite was small, but it was larger than that between ferrite and other adjacent structures.

As shown in Figure 5, in the grain size analysis of EBSD, the average grain size was measured by the intercept method. The color in the figure represents the grain orientation distribution, and the different color boundaries represent the grain boundaries. The test results of grain size are given in Table 4. After QT and SR treatment, the average grain size decreased.

Figure 6a–c was obtained by magnifying the grain boundary orientation pattern five times. Line A and B calibrate the orientation between cementite and ferrite and adjacent ferrite. In order to further analyze the grain formation mechanism of the samples in three states, the orientation of three adjacent grains was extracted by Channel 5 software. The three-dimensional display function of the software was used to simulate the three-dimensional images of these grain orientations, so as to understand the grain orientation characteristics more intuitively, as shown in Figure 6a–c, where a and b represent ferrite and c represents cementite.

Grain misorientation and grain orientation stereograms of specimens in different states are shown in Figure 6. Grain boundaries with adjacent grain orientation differences greater than 15° are defined as large-angle grain boundaries (LAGBs), and grain boundaries between 2° and 15° are defined as small-angle grain boundaries (SAGBs) [24]. It was obvious that the orientation difference between cementite and ferrite grains was less than 15°, which was a SAGB, and grain orientation was analogous. Since the orientation of adjacent grains on both sides of the SAGB was exactly the same, more SAGBs determined the smaller dislocation energy. The grain boundary energy was low, thus showing stronger resistance to intergranular corrosion [25]. The misorientation between adjacent ferrite grains was greater than 15°, which was a LAGB. The LAGB means that the dislocation between grains was large, and the larger the LAGBs between grains, the higher the dislocation of the specimen was proved. The LAGBs mean high-energy grain boundaries, which can slow down the propagation of brittle fracture cracks [26]. The distribution of the grain misorientation of the specimens after QT and SR treatment is illustrated in Figure 6b,c. Obviously, the content of SAGBs and LAGBs after heat treatment was more than that of as-deposited specimens. Grain boundary misorientation distributions of specimens in different states are shown in Figure 7. It is not difficult to see that the specimens in different states were mainly SAGBs. The SAGB content of the deposited specimens was 58%, and the LAGB content was 15%. After QT and SR, more of the finer grain SAGBs and LAGBs were obtained than those in the as-deposited specimens. The content of SAGB of QT specimens was 17% higher than that of the as-deposited state, and the content of LAGB was 2% higher than that of the as-deposited state. The LAGB content of SR specimens was 18%, and the SAGB content exhibited no significant change compared with the as-deposited state. The results indicated that in the range of SAGB, the interface energy between grain boundaries increased with the increase of the angle. When the grain boundary misorientation was 15°, the interface energy reached a peak, and then the interface energy did not change with the change of the grain boundary angle. At last stage of the crack propagation, the morphology of the microstructure, related to the spatial distribution of SAGBs and LAGBs,is expected to play an important role [27]. After QT treatment, the content of SAGB increased with the formation of polygonal ferrite, which enhanced its ability to resist intergranular corrosion.

### 3.3. Microhardness Test Results and Analysis

Figure 8 shows the hardness distribution of the cross-section of each test specimen in a different state. During the SLM forming process, they are under repeated thermal cycles. The internal structure of the specimens underwent a certain degree of tempering, which caused the hardness of the as-deposited specimens to gradually decrease from the surface layer to the base layer. In the QT process, high-temperature tempering led to complete decomposition of martensite so that the structure after quenching was converted into a tempering sorbite structure. At this time, the strength and hardness of the specimens decreased. After the specimens were subjected to SR, internal residual stress was released during the heating and holding process. The elastic strain field weakened or disappeared, and the hardness decreased.

### 3.4. Tensile Property Test Results and Analysis

Figure 9 displays the three-state tensile fracture morphology. It can be observed in the morphology of the tensile fracture that the specimens in the three states were all ductile fractures. A small amount of tear ridges can be observed. The appearance of tear ridges is another manifestation of the higher elongation of alloy steel.

As shown in Figure 10, after QT treatment, the quenched martensite decomposed due to high-temperature tempering. The carbon content of martensite decreased and the strength of steel decreased. From the EBSD analysis results, it can be seen that the grain size became smaller after QT and SR treatment. LAGBs effectively improved the plasticity of steel materials. Because of the deformation process, the length of the {110} slip surface determined the effective distance of dislocation slip and the formation of plugging [28,29]. The shorter the equivalent length of the {110} plane, the shorter the effective slip distance of the dislocation and less likely it was to form stress concentration [30]. Therefore, analysis from the perspective of the slip surface misorientation is that under conditions when both the ferrite and bainite microstructures were refined, the plasticity of the QT specimen was improved. On the other hand, the finer the grains, the more grains that may slip, so deformation can be dispersed in more grains. Therefore, plasticity and toughness were also improved [31].

### 3.5. High-Temperature Friction and Wear Test Results and Analysis

Table 5 shows the high-temperature friction and wear performance test results of specimens in different conditions. Figure 11a,b is respectively the high-temperature friction abrasion weight histogram and the friction coefficient curve of the as-deposited state, QT state and SR state.

Figure 11 shows the relationship curves of friction coefficient with time and the weight loss under three different conditions. The friction indicates that there was a greater contact stress between the friction pairs. A large friction coefficient indicates that there is a greater contact stress between the friction pairs, so the friction coefficient indirectly reflects the lubrication of the contact area of the friction pair or the formation of a stable friction protective film. It can be observed in Figure 11a that the weight loss of the as-deposited specimens was the largest, about 30 times that of the QT specimens, and twice that of the SR specimens. As shown in Figure 11b, during the pre-wear period during the first two minutes of wear, the friction coefficient curve rose rapidly. The friction coefficient of the QT specimens was slightly lower than that of the SR specimens.

Figure 12 shows the high-temperature friction and wear morphology of samples in different states. The wear morphology of the samples in the three states is similar. Clear furrows, spalling pits and oxide layers can be seen in the wear topography. It shows that there was mainly abrasive wear, adhesive wear and oxidation wear between wheel and rail materials. When the surface of the specimens was subjected to stress, plastic deformation occurred. When the contact stress exceeded the limit, fatigue cracks appeared on the surface of the specimen. Under the continuous action of load, cracks propagated gradually, and spalling pits formed on the surface of the material under the action of shear stress [32]. EBSD analysis shows that the grain size became smaller after heat treatment. The large-angle grain boundaries between grains increased, and the large-angle grain boundaries mean that the dislocations between grains was larger. It is proved that the samples had a high dislocation and the dislocation could be wound at the carbide. As a result, the movement of dislocations was hindered, and the ability of anti-deformation was improved [33,34]. The large-angle grain boundary means a high-energy grain boundary, which can slow down crack propagation [26], which is beneficial to the improvement of wear resistance. It can be seen from Figure 12 that the surface abrasion loss of the as-deposited specimens was the largest, and the QT specimens had the smallest abrasion loss. Therefore, the friction and wear performance of the QT specimens and the pin specimens were better matched. After QT treatment, a stable friction coefficient and good wear performance can be obtained.

### 3.6. Thermal Fatigue Test Results and Analysis

Figure 13 indicates the shape of the notch after the thermal fatigue test. Various degrees of oxidation pits and some reticular cracks appeared on the surface of the specimens. The surface of the as-deposited specimens had the most serious oxidation, with more spot-like lumps; the crack was longer, thicker, and deeper, and the crack length was 473.3 μm. The degree of oxidation of the SR specimens was lighter than that of the as-deposited state, and the crack length was 375.6 μm. The surface of the QT specimens had the lightest oxidation, and the crack length was 364.7 μm lower than that of the as-deposited state and the SR state. Due to the increase of the content of SAGB and the LAGB after heat treatment, it showed stronger resistance to intergranular corrosion and the ability to hinder crack growth [25,26]. Therefore, specimens after QT and SR had better oxidation resistance and thermal fatigue properties. Although the strength of the as-deposited specimens was higher, the plasticity and toughness were lower. The stability of microstructure in the process of the thermal cycle was also poor, so the thermal fatigue resistance was low. After heat treatment, the strength and plasticity of the specimens was well matched, and the stability of the microstructure was also higher, so the thermal fatigue resistance performance was high. After QT, with the formation of polygonal ferrite, the SAGB content also increased, and the resistance of the specimens to intergranular corrosion was stronger, so thermal fatigue performance of QT samples was better.

## 4. Conclusions

In this paper, the specimens of 24CrNiMo alloy steel were prepared by SLM with optimized process parameters. The related experiments were carried out, and the following conclusions were obtained:(1)The specimens of 24CrNiMo alloy steel were prepared by SLM with optimized process parameters. The density of the specimens approached 99.78%, and the tensile strength and yield strength of the as-deposited samples were 1199 MPa and 1053 MPa, respectively. The microstructure observation showed that the as-deposited samples were made up of ferrite, granular bainite, a small amount of cementite and residual austenite.(2)After QT and SR treatment, the elongation rate after fracture was higher than that of the deposited state, and was improved. After QT treatment, a tempered sorbite structure was obtained, and the microstructure of the SR specimens had no obvious change compared with the as-deposited state. As the internal residual stress of the specimens after QT and SR was released during the heating and holding processes, the elastic strain field weakened or disappeared, which led to the decrease of hardness.(3)The EBSD analysis results showed that the grain misorientation between cementite and ferrite was relatively small, while the grain misorientation between ferrite and ferrite was relatively large. The SAGB content of the as-deposited samples was 58%, and LAGB content was 15%. After QT and SR, more of the finer grains, SAGBs and LAGBs were obtained than those in the as-deposited specimens. The LAGB content of SR specimens was 18%, and the SAGB content exhibited no significant change compared with the as-deposited state. The SAGB content of the QT specimens was 17% higher than that of the deposited state, and the content of LAGB was increased by 2% compared with the deposited state, showing stronger resistance to intergranular corrosion performance and ability to hinder the crack propagation.(4)Compared with the as-deposited state, heat treatment effectively improved the high-temperature friction and wear performance and thermal fatigue performance of 24CrNiMo alloy steel. Especially after QT, with the formation of polygonal ferrite, the content of SAGB and LAGB increased, while the high-temperature friction and wear performance test and thermal fatigue performance also increased.

## Figures and Tables

**Figure 1 materials-14-00631-f001:**
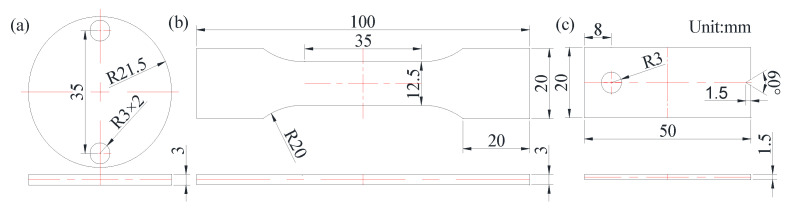
Specimen size: (**a**) high-temperature friction and wear; (**b**) tensile; (**c**) thermal fatigue.

**Figure 2 materials-14-00631-f002:**
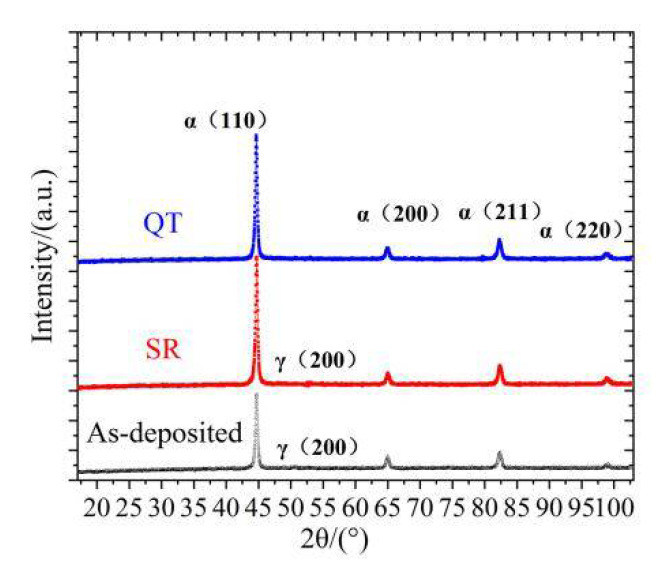
XRD patterns of different states.

**Figure 3 materials-14-00631-f003:**
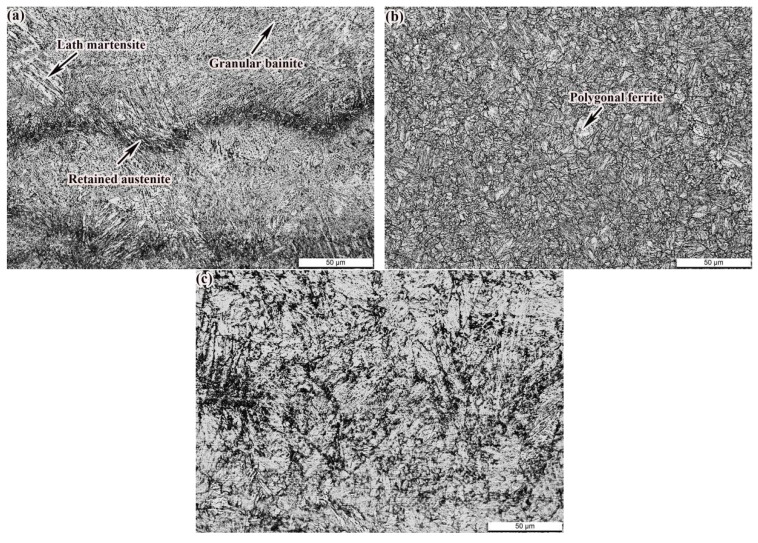
Microstructure in different states: (**a**) as-deposited; (**b**) quenched and tempered (QT); (**c**) and stress-relief annealed (SR).

**Figure 4 materials-14-00631-f004:**
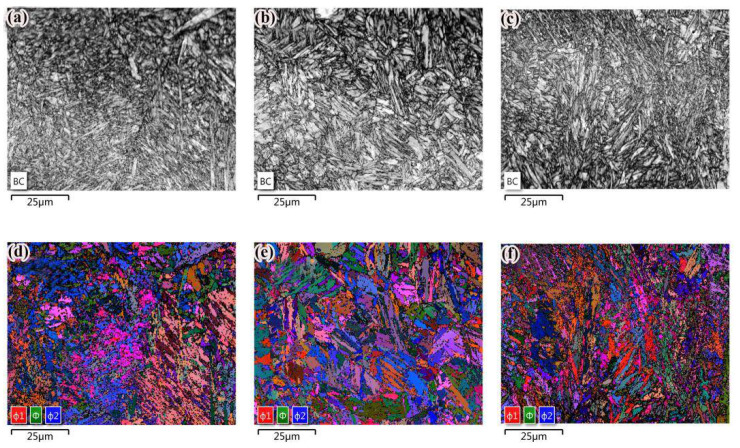
Analysis of phase electron backscattered scattering detection (EBSD): (**a**–**c**) represent the quality imaging images of as-deposited, QT and SR specimens; (**d**–**f**) represent the Euler angular orientation reconstruction of as-deposited, QT and SR specimens.

**Figure 5 materials-14-00631-f005:**
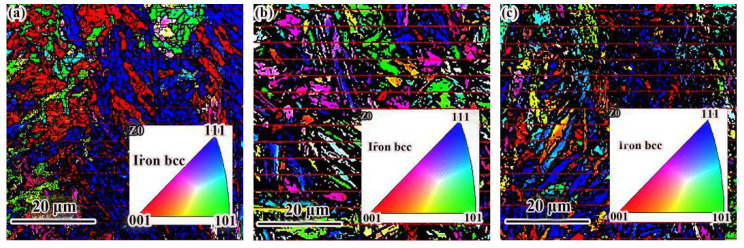
Grain size and grain boundary orientation: (**a**) as-deposited; (**b**) QT; (**c**) SR.

**Figure 6 materials-14-00631-f006:**
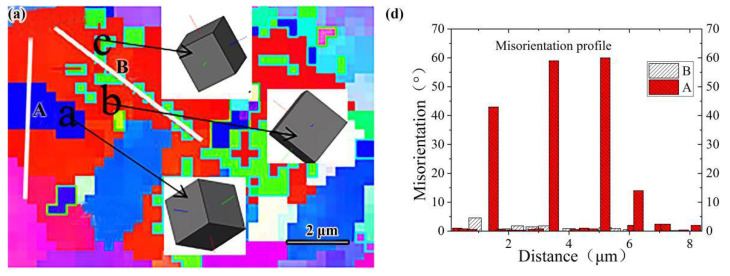
Grain misorientation and grain orientation stereogram: (**a**–**c**) represent the magnified view of grain boundary orientation of as-deposited, QT and SR specimens; (**d**–**f**) represent the grain misorientation of the as-deposited, QT and SR specimens.

**Figure 7 materials-14-00631-f007:**
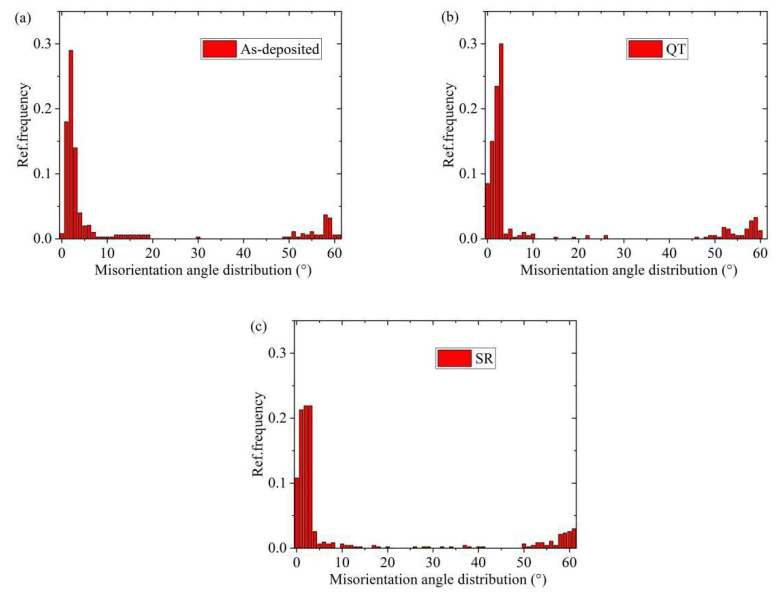
The misorientation angle distribution histograms of specimens: (**a**) as-deposited; (**b**) QT; (**c**) SR.

**Figure 8 materials-14-00631-f008:**
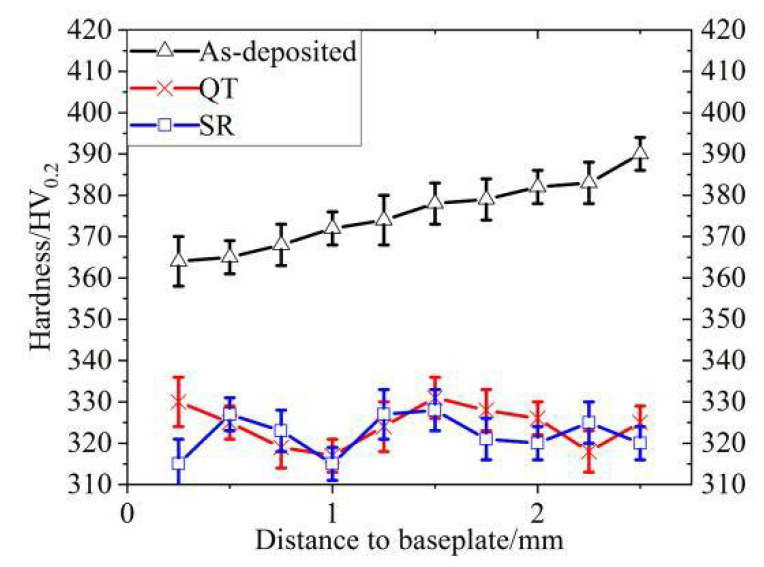
Hardness of samples in different states.

**Figure 9 materials-14-00631-f009:**
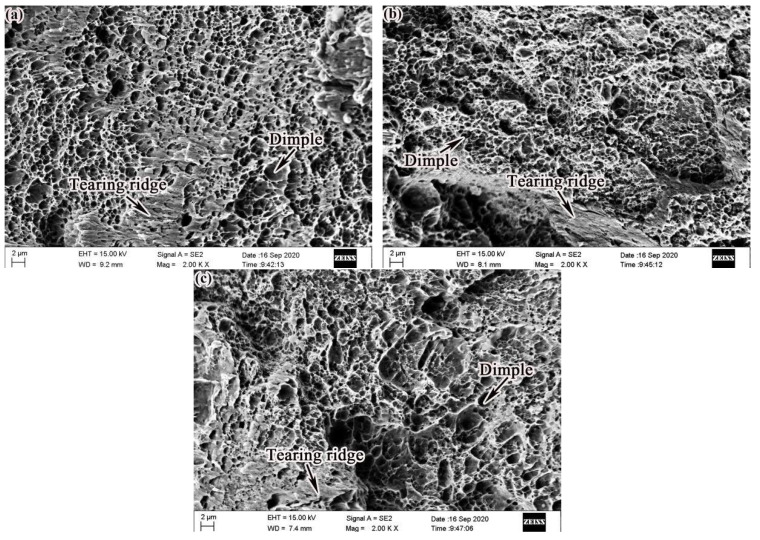
Tensile fracture morphology: (**a**) as-deposited; (**b**) QT; (**c**) SR.

**Figure 10 materials-14-00631-f010:**
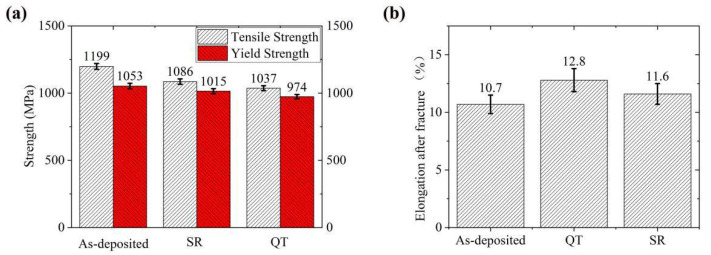
Tensile test results: (**a**) tensile strength and yield strength; (**b**) elongation after fracture.

**Figure 11 materials-14-00631-f011:**
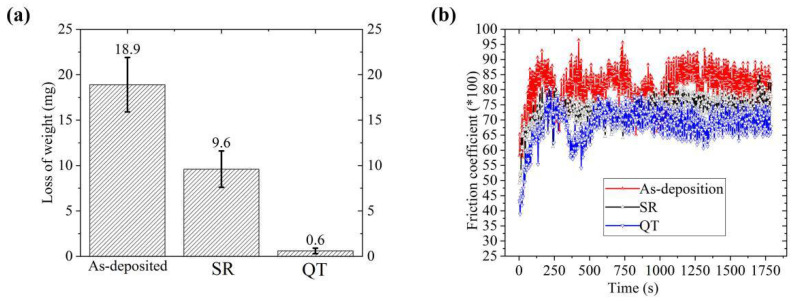
Weight and coefficient of friction at high-temperature: (**a**) the weight lost by friction; (**b**) friction coefficients.

**Figure 12 materials-14-00631-f012:**
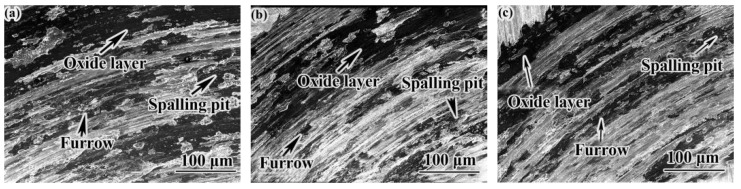
High temperature friction and wear morphologies of samples in different states: (**a**) as-deposited; (**b**) QT; (**c**) SR.

**Figure 13 materials-14-00631-f013:**
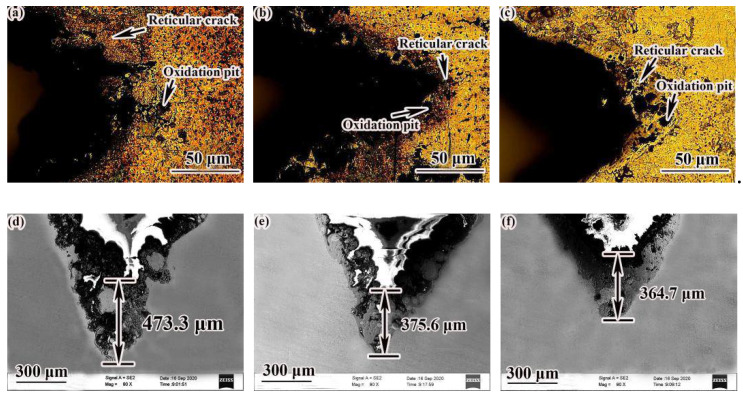
Crack morphology of samples in different states after thermal fatigue test: (**a**,**d**) as-deposited; (**b**,**e**) SR; (**c**,**f**) QT.

**Table 1 materials-14-00631-t001:** Chemical composition of 24CrNiMo alloy steel and 30CrNiMo steel.

Materials	FE	C	MN	NI	MO	SI	CR	S
24CRNIMO	Bal	0.23	0.72	1.81	0.47	0.21	1.12	0.0031
30CRNIMO	Bal	0.31	0.68	0.02	0.16	0.26	0.98	0.004

**Table 2 materials-14-00631-t002:** Heat treatment process of 24CrNiMo alloy steel sample.

Sample State	Heat Treatment Parameters
As-deposited	-
QT	860 °C × 30 min/OC + 600 °C × 40 min/AC
SR	600 °C × 120 min/FC

**Table 3 materials-14-00631-t003:** Test parameters of high-temperature friction wear and thermal fatigue.

Test Methods	Test Parameters			
High-temperature friction and wear	Time/min	Temperature/°C	Rotational speed/r·min^−1^	Load/N
	30	500	300	100
Thermal fatigue	Number of cycles	Maximum temperature/°C	Lowest temperature/°C	Coolant
	2000	550	25	Water

**Table 4 materials-14-00631-t004:** Grain sizes in different states.

Sample State	Average Grain Size/μm	
	Iron bcc	Fe_3_C
As-deposited	1.8642	0.4342
QT	1.5918	0.3417
SR	1.2609	0.3402

**Table 5 materials-14-00631-t005:** High-temperature friction and wear performance test results.

Sample State	Lose Weight (mg)	Friction Coefficient
As-deposited	18.9	0.80–0.87
SR	9.6	0.7–0.75
QT	0.6	0.64–0.7

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
