# Peer review of "Effect of Heat Treatment on Microstructure and Properties of 24CrNiMo Alloy Steel Formed by Selective Laser Melting (SLM)"

_materials, 2021, doi:10.3390/ma14030631_

Round 1
Reviewer 1 Report
Comments on improving the manuscript:
1) the abstract and conclusions should specify the novelty of the study;
2) in the abstract, align the designations of strength and yield strength (remove the comma) and be sure to Supplement the quantitative indicators of the effect of heat treatment on the properties of 24CrNiMo Alloy Steel by SLM forming (EBSD analysis results, grain misorientation, the content of small-angle and large-angle grain, friction and wear properties and thermal fatigue properties);
3) in the keywords to exclude a phrase;
4) figure 1 should be improved: in the sample drawing, add center lines, make the diameter symbol correct, and exclude the intersection of numbers with lines;
5) in the text of clause 3.1.1 enter the intensity values for As-deposited, QT and SR;
6) add missing figure 5;
7) eliminate the merging of the experimental points in figure 7 ("o "replace" x") and 10b (black color replace with gray);
8) it is necessary to add to the text of 3.6 the measurement results: 473.3 microns, 375.6 microns and 364.7 microns;
9) the conclusion should also Supplement the quantitative indicators of the effect of heat treatment on the properties of 24CrNiMo Alloy Steel by SLM forming (EBSD analysis results, grain misorientation, the content of small-angle and large-angle grain, friction and wear properties and thermal fatigue properties) and show the prospect of further research.
Author Response
Response to Reviewer 1 Comments
Point 1: the abstract and conclusions should specify the novelty of the study;
Point 2: in the abstract, align the designations of strength and yield strength (remove the comma) and be sure to Supplement the quantitative indicators of the effect of heat treatment on the properties of 24CrNiMo Alloy Steel by SLM forming (EBSD analysis results, grain misorientation, the content of small-angle and large-angle grain, friction and wear properties and thermal fatigue properties);
Point 3: in the keywords to exclude a phrase;
Point 4: figure 1 should be improved: in the sample drawing, add center lines, make the diameter symbol correct, and exclude the intersection of numbers with lines;
Point 5: in the text of clause 3.1.1 enter the intensity values for As-deposited, QT and SR;
Point 6: add missing figure 5;
Point 7: eliminate the merging of the experimental points in figure 7 ("o "replace" x") and 10b (black color replace with gray);
Point 8: it is necessary to add to the text of 3.6 the measurement results: 473.3 microns, 375.6 microns and 364.7 microns;
Point 9: the conclusion should also Supplement the quantitative indicators of the effect of heat treatment on the properties of 24CrNiMo Alloy Steel by SLM forming (EBSD analysis results, grain misorientation, the content of small-angle and large-angle grain, friction and wear properties and thermal fatigue properties) and show the prospect of further research.
Response 1: The abstract and conclusions have explained the novelty of the study, such as the relationship between the content of small-angle grain boundaries and the content of large-angle grain boundaries and performance.
Response 2:The comma has been removed from the name of tensile strength and yield strength, and the quantitative indicators of the effect of heat treatment on the properties of 24CrNiMo Alloy Steel by SLM forming has been added (EBSD analysis results, poor grain orientation, small and large angle grain content, friction and wear properties and thermal fatigue properties).
Response 3:A phrase has been excluded from the keyword.
Response 4: In figure 1, the center lines has been added, the diameter symbol has been modified, the radius symbol has been replaced, and the intersection of numbers and straight lines has been excluded.
Response 5: The intensity values of deposited state, QT and SR have been entered in the text of clause 3.1.1 .
Response 6: The missing figure 5 has been added.
Response 7: I have been eliminated the merging of the experimental points in figure 7 ("o "replace" x") and 10b (black color replace with gray);
Response 8: The measurement results have been added to the text of 3.6.: 473.3 microns, 375.6 microns and 364.7 microns;
Response 9: Conclusion the quantitative indicators of the effect of the effect of heat treatment on the properties of 24CrNiMo Alloy Steel by SLM forming (EBSD analysis results, grain misorientation, the content of small-angle and large-angle grain, friction and wear properties and thermal fatigue properties) have been supplemented, and the prospect of further research remains to be studied.

Reviewer 2 Report
Dear Authors.
Please use proper Material journal template for preparing your paper. It looks really messy and unarranged.
Introduction should be rewritten with more related papers.
Even though the paper has a reasonable results, it is not clearly presented in the article. It looks really messy.
Please check the pdf file for comments.
I will check the document once it is properly updated.
Thank You.

Author Response
Response to Reviewer 2 Comments
Point 1: Please use proper Material journal template for preparing your paper. It looks really messy and unarranged.
Point 2:Introduction should be rewritten with more related papers.
Point 3:Even though the paper has a reasonable results, it is not clearly presented in the article. It looks really messy.
Response 1: I have used the proper Material journal template template to prepare the paper.
Response 2:The introduction has been rewritten with more related papers.
Response 3: Reasonable results have been clearly presented in this article.
This revision edited the language and format of the article. Checked grammar, spelling, punctuation, etc.

Round 2
Reviewer 1 Report
The changes made significantly improved the manuscript.
Author Response
Point 1: The changes made significantly improved the manuscript.
Response 1: Thank you for your suggestion. I have made a small revision. If you have any questions, please contact me.

Reviewer 2 Report
The research idea of the paper is good and the findings of the research study proves the proposed idea clearly. However there are some mistakes which i want to address and paper can be published after making these changes •The paper content are good but poorly written and need an advance level of editing; as many sentences doesn’t provide the idea what the author wants to address at these lines. For example (line 107-108), please rearrange these type of sentences, also the connection between each sentences is very poor which disturbs the flow of readings, I advice you to increase the paragraph length while also add connection words like (however, furthermore, moreover etc). •Also there is a spacing problem in many sentences start which I highlighted with yellow color in the pdf file, so remove that and also the title of figure are not centered (Fig.5, 6). Also increase the size of figure (2&7), which is hard to read for the reader in the standard size, misalignment of Figure 9(a&b) in vertical direction. • Also please check the template one more time because I noticed the title location of each sections as well as font of table text to be out of template. •Introduction section is not in proper format location, also in Fig. 1, some texts are bold and some are normal.

Author Response
Point 1: •The paper content are good but poorly written and need an advance level of editing; as many sentences doesn’t provide the idea what the author wants to address at these lines. For example (line 107-108), please rearrange these type of sentences, also the connection between each sentences is very poor which disturbs the flow of readings, I advice you to increase the paragraph length while also add connection words like (however, furthermore, moreover etc).
Point 2: •Also there is a spacing problem in many sentences start which I highlighted with yellow color in the pdf file, so remove that and also the title of figure are not centered (Fig.5, 6). Also increase the size of figure (2&7), which is hard to read for the reader in the standard size, misalignment of Figure 9(a&b) in vertical direction.
Point 3: •Also please check the template one more time because I noticed the title location of each sections as well as font of table text to be out of template.
Point 4: •Introduction section is not in proper format location, also in Fig. 1, some texts are bold and some are normal.
Response 1: For (lines 107-108), due to the inconsistency between the PDF file you marked and the manuscript you uploaded the second time, I have rearranged these types of sentences (for example, lines 107-108) the last time I revised the manuscript, and the connection between each sentence has been adjusted appropriately. At the same time, I also increased the length of the paragraph.
Response 2: I have corrected the spacing problem in many opening sentences. The format of the graphic title in figures 5 and 6 has been modified. Figure 7 I have increased the size, but for figure 2, there is no size, this is the XRD diagram, there is no way to increase the size. Figure 9 (a&b) has been aligned vertically.
Response 3&4: I have checked the template again and made changes to the format of the introduction section, as well as to figure 1. In addition, as for the size of figure 3, it is impossible to modify it because of the size that comes with it when the photo is taken. If you have any questions, please contact me in time.
